# Sensorless Speed Estimation of Induction Motors through Signal Analysis Based on Chaos Using Density of Maxima

**DOI:** 10.3390/e26050361

**Published:** 2024-04-25

**Authors:** Marlio Antonio Silva, Jose Anselmo Lucena-Junior, Julio Cesar da Silva, Francisco Antonio Belo, Abel Cavalcante Lima-Filho, Jorge Gabriel Gomes de Souza Ramos, Romulo Camara, Alisson Brito

**Affiliations:** 1Graduate Program in Mechanical Engineering Department, Federal University of Paraiba (UFPB), João Pessoa 58051-900, Brazil; marlio.silva@academico.ufpb.br (M.A.S.); anselmojr_mec_ufpb@hotmail.com (J.A.L.-J.); julio.cesar3@academico.ufpb.br (J.C.d.S.); belo@cear.ufpb.br (F.A.B.); abelima@gmail.com (A.C.L.-F.); 2Graduate Program in Physics, Federal University of Paraiba (UFPB), João Pessoa 58051-900, Brazil; jorgegabrielramos@gmail.com; 3Graduate Program in Informatics, Federal University of Paraiba (UFPB), João Pessoa 58051-900, Brazil; romulo.camara@ci.ufpb.br

**Keywords:** chaos, stochastic process, time series, three-phase induction motors

## Abstract

Three-phase induction motors are widely used in various industrial sectors and are responsible for a significant portion of the total electrical energy consumed. To ensure their efficient operation, it is necessary to apply control systems with specific algorithms able to estimate rotation speed accurately and with an adequate response time. However, the angular speed sensors used in induction motors are generally expensive and unreliable, and they may be unsuitable for use in hostile environments. This paper presents an algorithm for speed estimation in three-phase induction motors using the chaotic variable of maximum density. The technique used in this work analyzes the current signals from the motor power supply without invasive sensors on its structure. The results show that speed estimation is achieved with a response time lower than that obtained by classical techniques based on the Fourier Transform. This technique allows for the provision of motor shaft speed values when operated under variable load.

## 1. Introduction

Three-phase induction motors (TIMs) are widely used in various industrial environments mainly due to their characteristics: versatility, low cost, robustness, and high efficiency. However, the intense use of TIMs is also responsible for a significant portion of the total electricity consumption in the industry, accounting for about 68% [1]. Therefore, continuous observation is necessary to improve energy efficiency in their applications. For this reason, several studies have presented speed control, fault diagnosis, and estimation methods in TIMs, aiming to reduce their electrical energy consumption without compromising their dynamic performance.

Conceived initially as constant-speed electric motors, TIMs dramatically expanded their applications through the advent of Variable-Frequency Drives (VFDs), which, in turn, made it possible to control the operating parameters of these electric machines, such as speed and torque. As an example of the potential for efficient energy use in this equipment, we can observe that the law of similarity of rotors describes energy consumption in fluid pumping applications by centrifugal pumps. That is, parameters such as power and torque vary proportionally with the cube and square of the speed, respectively [2]. There is a 30% speed reduction and a 66% decrease in power required for operation in this application.

TIMs are also known to be nonlinear dynamic systems. Their parameters, such as resistance, current, and inductance, vary with time and operating modes. The accuracy of the speed estimation based on these parameters strongly depends on the need to fine-tune these parameters included in the algorithm used. Consequently, any incompatibility of the parameters can imply instability of the frequency inverter and errors in the speed estimation [2].

Sensorless techniques are highlighted, with their main advantages being a reduction in the complexity of the equipment structure, low cost, controller simplification, elimination of the need for cables, better noise immunity, increased reliability, and lower maintenance requirements. Furthermore, the angular speed sensors used in induction motors are generally expensive and unreliable [3]. Likewise, hostile environments usually require induction motors to operate without mechanical sensors [4,5]. Speed estimation can generally be performed using two approaches: direct measurement by a harmonics injection and through the supply of TIM currents or voltage signals. These approaches use specific algorithms that replace the rotor position sensor, thus eliminating the need to install a mechanical position sensor on the motor shaft [6,7].

In addition to the classical techniques, we can observe the nature of chaotic signals and how they have helpful information for speed estimation. Chaotic signals can be present in nonlinear dynamic systems and are characterized through several techniques in the literature. Several electrical systems exhibit chaotic signals with distinct features, including their apparent randomness, stemming from their non-repetitive nature and a significant reliance on their initial conditions. For these reasons, studies of chaotic signals, especially in electrical systems, become interesting, as these signals have essential information that could, at first glance, be mistakenly interpreted as noise.

Within the concept of a chaos analysis, the SAC-DM (Signal Analysis based on Chaos using Density of Maxima) technique has been extensively explored in analyzing the behavior of brushless direct-current (BLDC) motors [8,9,10], internal combustion engines (ICEs) [11], and induction motors (IMs) [12], mainly due to its high sensitivity to small changes in a motor’s behavior, which can indicate anything from incipient failures to variations in the shaft rotation speed, as will be seen for induction motors for the first time in this work.

In the next section, we present the state of the art related to the topic of this article, which identifies the originality of the proposed technique and its main contributions.

## 2. State of the Art

Table 1 shows a selection of recent studies that have utilized the armature current from three-phase induction motors to estimate shaft rotation speeds, demonstrating a range of signal processing techniques. These techniques vary primarily by their signal processing domain, leading to differences in estimation response times and accuracy, as measured by relative percentage errors.

It can also be seen that chaos theory was used for the first time to estimate speed in induction motors, using a hybrid approach (time/frequency), obtaining results compatible with those found in the state of the art, as well as eliminating the use of engine nameplate data to implement the technique, contrary to what happens in works based on the engine model or equivalent circuit.

This paper aims to present an algorithm for speed estimation through current signals at TIM terminals, using the chaotic component of the resulting signal. Among the main innovation items and contributions of this work, we can highlight the following:This work marks the first instance of applying chaos theory to predict the rotational speed of induction motor shafts.The presented technique is based on a hybrid, one-dimensional, time domain approach using the frequency of occurrence of peaks in the current signal, providing significant effectiveness in the estimation and a shorter response time than most related work, as well as eliminating the use of engine nameplate data to implement the technique.

## 3. Signal Analysis Based on Chaos Using Density of Maxima (SAC-DM)

The methodological innovation introduced by the authors in [22] facilitates the robust identification of chaotic dynamics across diverse systems, predicated upon a fundamental time-series analysis of measurable system parameters. This framework has been meticulously validated through a rigorous application of the Hamming distance metric, detailed comprehensively in [23]. Importantly, this framework not only facilitates the accurate identification of chaos but also introduces a new method for measuring the correlation length within chaotic dynamics—a task that has traditionally required a significant amount of data and time resources. This challenge is elegantly surmounted by a straightforward measure of the density of maxima, calculated as the ratio of maxima occurrences within a specified time interval. Consequently, this technique transcends chaos identification, serving as the cornerstone for developing signal processing methodologies rooted in variable chaotic maximum density, collectively denominated as Signal Analysis based on Chaos using Density of Maxima (SAC-DM) [9,10]. Through the proposed analytical framework, characterized by the minimal signal sample in time denoted as $q_{i}\left(t\right)$, the chaotic behavior within the system under scrutiny, exemplified herein by the TIM armature current signal, is meticulously delineated.

To render this paper self-contained, we shall elucidate the fundamental concept underlying the computation of correlation length, typically requiring substantial data, via SAC-DM, which necessitates only scant data. Let us consider the interval [t,t+δt]. The signal sample $q_{i}\left(t\right)$ evolves and oscillates, yielding a local maximum. For sufficiently small δt, it follows that the first derivative at time $q_{i}^{\prime }\left(t\right)>0$ and $q_{i}^{\prime }\left(t+\delta t\right)<0$, ensuring $-q_{i}^{\prime \prime }\left(t\right)\delta t>q_{i}^{\prime }\left(t\right)>0$. The joint probability $P(q_{i}^{\prime },q_{i}^{\prime \prime })$ serves to compute the average maximum density, denoted as $\langle \rho_{i}\rangle$. Therefore, the likelihood of finding a maximum within the interval [t,t+δt] is directly proportional to the integral function that covers the specified interval, given by
(1)$$\begin{array}{ccc} \langle \rho_{i}\rangle & \equiv & \frac{1}{\delta t}\int_{-\infty }^{0}dq_{i}^{\prime \prime }\int_{0}^{-q_{i}^{\prime \prime }\delta t}dq_{i}^{\prime }P\left(q_{i}^{\prime },q_{i}^{\prime \prime }\right)\\ & = & \int_{-\infty }^{0}dq_{i}^{\prime \prime }q_{i}^{\prime \prime }P\left(0,q_{i}^{\prime \prime }\right) \end{array}$$

The mean values of the terms $q_{i}^{\prime }$ and $q_{i}^{\prime \prime }$ tend to be zero due to the statistical properties of the mean number of maxima, which are invariant under time translation. Through the smallest instances of $q_{i}^{\prime }$ and $q_{i}^{\prime \prime }$, the properties of $P(q_{i}^{\prime },q_{i}^{\prime \prime })$ can be achieved, and its variances are directly related to the correlation function:(2)$$C_{i}\left(\delta t\right)=\langle q_{i}\left(t+\delta t\right)q_{i}\left(t\right)\rangle$$

Deriving $C_{i}\left(\delta t\right)$, we obtain
(3)$$\begin{array}{ccc} \langle {q_{i}^{\prime }}^{2}\rangle & = & -\frac{d^{2}C_{i}\left(\delta t\right)}{d{\left(\delta t\right)}^{2}}{|}_{\delta t=0}\\ \langle {q_{i}^{\prime \prime }}^{2}\rangle & = & \frac{d^{4}C_{i}\left(\delta t\right)}{d{\left(\delta t\right)}^{4}}{|}_{\delta t=0} \end{array}$$

The joint probability distribution can be constructed using the maximum entropy principle for $q_{i}$ and its derivatives presented in the previous equations. After implementing algebraic calculations, the integration of $q_{i}$ leads to $P(q_{i}^{\prime },q_{i}^{\prime \prime })$’, shown as follows:(4)$$P\left(0,q_{i}^{\prime \prime }\right)=\frac{1}{2\pi }\frac{1}{\sqrt{\langle q_{i}^{\prime 2}\rangle \langle q_{i}^{\prime \prime 2}\rangle }}\exp\left(-\frac{1}{2}\frac{q_{i}^{\prime \prime 2}}{\langle q_{i}^{\prime \prime 2}\rangle }\right)$$

The expression depicted in Equation (Equation 4) provides a means to ascertain the density of maxima with the autocorrelation function. Equations (Equation 1)–(Equation 4) can be employed to derive $\rho_{i}$, after some algebraic manipulation, as follows:(5)$$\langle \rho_{i}\rangle =\frac{1}{2\pi }\sqrt{\frac{\langle q_{i}^{\prime \prime 2}\rangle }{\langle q_{i}^{\prime 2}\rangle }}=\frac{1}{2\pi }\sqrt{\frac{\frac{d^{4}C_{i}}{d{\left(\delta t\right)}^{4}}\left(0\right)}{-\frac{d^{2}C_{i}}{d{\left(\delta t\right)}^{2}}\left(0\right)}}$$

As conventionally recognized, the correlation length τ is deduced from the correlation function, typically interpreted as the width at half maximum. Employing the fitting function C(t)=cos(κt), we establish κτ=π/3. Consequently, from Equations (Equation 3) and (Equation 5), one deduces the conclusion that $\langle \rho_{i}\rangle =\kappa /\left(2\pi \right)$. This establishes a relationship between the density of maxima and the correlation length, expressed as $\tau =1/(6\langle \rho_{i}\rangle )$. Thus, we can readily infer the correlation length utilizing a simple measure provided by SAC-DM.

This framework offers a new quantity that appears from the chaotic behavior present in a stochastic system. Having proven the chaotic behavior of a mechanical system through the analysis of a signal emitted by it, Equation (Equation 6) can be inferred from time windows Δt, as has been proven in previous work [12]:(6)$$\langle \rho_{i}\rangle =\langle SAC-DM\rangle =\frac{\mathrm{Number}\;\mathrm{of}\;\mathrm{Peaks}}{\Delta t}$$

Starting from this section, the experimental chaotic component will be known as SAC-DM. This relation offers a simple way of estimating the speed of the studied system since it is verified through the results that, for each value of the TIM operation speed, there is a distinct range of values of the SAC-DM. In this way, it is possible to link the shaft rotation speed with the value obtained through the SAC-DM chaotic variable of the armature current signal of an induction motor, allowing for the rotation speed to be estimated in real time through the SAC-DM value. To achieve this, it is necessary to confirm the chaotic/deterministic behavior of the armature current signal, whose theoretical basis used for this purpose will be the symbol tree test and the 0-1 test for chaos, presented in the following sections.

## 4. Symbol Tree Test

Since the discovery of chaotic time series, researchers have been dedicated to discerning whether the signal acquired in an experiment exhibits chaotic or random characteristics. As previously mentioned, chaotic signals possess distinct features required for accurate classification. The approach outlined by [24] involves symbolic techniques to test for determinism in time series.

At a specific level within the symbolic tree, the behavior of the symbolic spectrum differs significantly between deterministic and stochastic time series. In the case of deterministic time series, the repeatability of the symbolic spectrum yields positive results, in contrast to what is observed in stochastic time series. These applications were carried out on simulated chaotic time series, such as the logistic map and the Henon map, as well as on stochastic time series, including Gaussian white noise.

The conversion of a time series $\left(x_{i}\right)$ of length *N* into a symbolic sequence $\left(S_{i}\right)$ is accomplished by subjecting it to a threshold function as follows:(7)$$\left\{x_{i}\right\}\to \left\{S_{i}\right\},\;i=1,2,\ldots ,N$$

Here, $S_{i}\in \left(0,1\right)$. Therefore, the threshold function is defined as follows:(8)$$\begin{array}{cc} \mathrm{If}\;x_{i}<\mathrm{median}\left(\left\{x_{i}\right\}\right), & S_{i}\equiv 0 \end{array}$$(9)$$\begin{array}{cc} \mathrm{If}\;x_{i}>\mathrm{median}\left(\left\{x_{i}\right\}\right), & S_{i}\equiv 1 \end{array}$$

The symbol tree is constructed from the sequence of symbols $S_{i}$, as presented in Figure 1.

The symbol tree is structured so that each term symbolizes the probability of a particular sequence, as denoted by its corresponding subscripts, within the symbolic sequence. For example, $P_{010}$ is the probability of observing the sequence 010 in the sequence of symbols. Each row of the symbol tree corresponds to a level, with the first row denoted as L=1, the second as L=2, and so on. In a binary system, each row presents probabilities equal to 2L, implying that, at a certain level, for example, L=2, there will be four different types of probabilities. These rows are defined as the symbol spectrum of level *L* [24].

In [25], the authors suggest that the symbol tree test begins by dividing a binary series of length *N* into subsets of length *l*. This division can be performed in two ways: first, by distributing into disjoint subsets of *l* and, second, by randomly selecting subsets of *l*.

The next step is the level *L* selection of the symbol tree. For each division of a binary series of length *l*, there will be l−(L−l) types of probabilities (referred to as “words” in the cited work). Therefore, the second element of one word will be the first element of the next word. It is recommended that each word be converted into decimal form to expedite the symbol spectrum test calculations. As an example of the steps for conducting the symbol spectrum test, consider a time series divided such that l=6. In this case, one of the divisions might be {0,1,0,0,1,0}, and if L=2 is chosen, the divisions would become {01,10,00,01,10}, which, in decimal form, would be {1,2,0,1,2}.

Deterministic series should exhibit a significant overlap in their spectra, unlike random series, where this characteristic will be sparse between one spectrum and the next [26].

## 5. The 0-1 Test for Chaos

The 0-1 chaos test, an algorithm designed to determine whether time series data exhibit chaotic behavior, was developed specifically for deterministic series, as outlined in [27]. The theoretical foundation of the 0-1 test is provided in [28], with a practical application guide offered in [29]. Opposite to the method for calculating the maximum Lyapunov exponent, this test can be directly applied to time series data without the need for phase-space reconstruction. This algorithm processes time series data as its input and yields a binary outcome, signifying whether the underlying dynamical system exhibits chaotic behavior. This test can be applied to any deterministic dynamical system, including ordinary and partial differential equations, and maps [27].

The 0-1 chaos test is performed as follows: consider a time series ϕ(j), for j=1,...,N. For c∈(0,π), the conversion variables are calculated as shown below:(10)$$\begin{array}{cc} p_{c}\left(n\right) & =\sum_{j=1}^{n}\phi \left(j\right)\cos\left(jc\right) \end{array}$$(11)$$\begin{array}{cc} q_{c}\left(n\right) & =\sum_{j=1}^{n}\phi \left(j\right)\sin\left(jc\right), \end{array}$$
where n=1,2,...,N.

The diffusive or non-diffusive behavior of the conversion variables $p_{c}$ and $q_{c}$ can be investigated by analyzing the mean square displacement defined by Mc(n). The tests performed in the study in [27] ensure that, if the dynamics are regular, this implies that the mean square displacement will be a limiting function of time, whereas if the dynamics are chaotic, $M_{c}\left(n\right)$ will grow linearly in time. Equation (Equation 12) shows the expression for the mean square displacement in terms of the conversion variables:(12)$$M_{c}\left(n\right)=\lim_{N\to \infty }\frac{1}{N}\sum_{j=1}^{N}\left\{{\left[p_{c}\left(j+n\right)-p_{c}\left(j\right)\right]}^{2}+{\left[q_{c}\left(j+n\right)-q_{c}\left(j\right)\right]}^{2}\right\}$$

The above equation requires the condition n ≪ N to be true, and this condition is guaranteed by calculating $M_{c}\left(n\right)$ only for $n\leq n_{\mathrm{cut}}$, where $n_{\mathrm{cut}}=N/10$ yields practical results [29].

Furthermore, the authors suggest that the chaos test is based on the growth of the value of $M_{c}\left(n\right)$ as a function of *n*. For each value of c∈(0,π), $M_{c}\left(n\right)$ takes the form of Equation (Equation 13), where V(c) is the slope adjustment term, and $V_{osc}\left(c,n\right)$ is the oscillatory term.
(13)$$M_{c}\left(n\right)=V\left(c\right)n+V_{osc}\left(c,n\right)+e\left(c,n\right),$$
where e(c,n) is the error term, and, if e(c,n)→0 as n→∞, it is given by
(14)$$V_{osc}\left(c,n\right)={\left(E\phi \right)}^{2}\left(1-\frac{1-\cos(nc)}{1-\cos(c)}\right)$$

In Equation (Equation 15), E[ϕ] is the expected value of the time series, given by
(15)$$E\left[\phi \right]=\frac{1}{N}\sum_{j=1}^{N}\phi_{j}$$

Without the error term e(c,n), shown in Equation (Equation 5), $M_{c}\left(n\right)$ takes the form of a cosine curve with slope V(c). It is important to note that the term V(c) is constant for a given value of *c*. According to [26], this slope characterizes the dynamics. To determine the slope, the subtraction of the term $V_{osc}$ from $M_{c}\left(n\right)$ is performed, creating a modified mean square displacement:(16)$$D_{c}\left(n\right)=M_{c}\left(n\right)-V_{osc}\left(c,n\right)$$

Finally, the calculation determines the asymptotic growth rate $K_{c}$ of the modified mean square displacement $D_{c}$. In [29], the authors present two methods for determining the term $K_{c}$, namely, the regression method and the correlation method. This work uses the correlation method presented in the cited work.

The asymptotic growth rate is the correlation coefficient of the following vectors:(17)$$\begin{array}{cc} \xi & =(1,2,\ldots ,n_{\mathrm{cut}}) \end{array}$$(18)$$\begin{array}{cc} \Delta & =(D_{c}\left(1\right),D_{c}\left(2\right),\ldots ,D_{c}\left(n_{\mathrm{cut}}\right)) \end{array}$$

Hence, given two vectors, *x* and *y*, of length *q*, the covariance and variance are defined as
(19)$$\begin{array}{cc} \mathrm{cov}(x,y) & =\frac{1}{q}\sum_{j=1}^{q}\left(x_{j}-\bar{x}\right)\left(y_{j}-\bar{y}\right) \end{array}$$
(20)$$\begin{array}{cc} \bar{x} & =\frac{1}{q}\sum_{j=1}^{q}x_{j} \end{array}$$
(21)$$\begin{array}{cc} \mathrm{var}(x) & =\mathrm{cov}(x,x) \end{array}$$

And then, for the correlation coefficient:(22)$$K_{c}=\mathrm{corr}\left(\xi ,\Delta \right)=\frac{\mathrm{cov}(\xi ,\Delta )}{\sqrt{\mathrm{var}(\xi )\mathrm{var}(\Delta )}}\in \left[-1,1\right]$$

The term $K_{c}$ measures the strength of the correlation of $D_{c}$ with linear growth, and, practically, the correlation method outperforms the regression method. Its values are $K_{c}=1$ for chaotic dynamics and $K_{c}=0$ for regular dynamics.

## 6. Multiresolution Analysis (MRA)

Multiresolution is an algorithm that applies the discrete wavelet transform using a multistage filter bank, with the wavelet function Ψ(t) used as a low-pass filter and the dual of this function used as a high-pass filter. In multiresolution theory, an original discrete signal is decomposed into two components, A1 (signal approximation) and D1 (signal detail), by a low-pass filter and a high-pass filter, respectively. For the second level, the approximation A1 is decomposed into another approximation, A2, and a detail, D2; this procedure is repeated for the third level, the fourth, and so on.

In this work, the signals resulting from the multiplication of the TIM armature current signals are decomposed into one approximation and seven details, with each signal primarily composing a specific frequency range, which depends on the acquisition rate used. In this work, the acquisition rate was 30 thousand samples per second, and the distribution of frequencies between the decomposed signals is shown in Table 2.

To isolate the component of the signal that causes chaotic behavior, the oscillatory component in the frequency range of 0 Hz to 117 Hz (represented by approximation A7 in Figure 2) needs to be eliminated. After this step, the remaining details will only process the desired component of the signal.

After the MRA decomposition step of the resulting signal from the multiplication of phases $i_{a}\times i_{b}$, it is necessary to confirm whether the signal is chaotic. For this purpose, determinism tests (symbol tree test) and chaos tests (0-1 test) were conducted. The test results can be found in Section 8.

## 7. Methodology

A set of equipment that integrates the test bench allows for the application of controlled loads on a TIM (shown in Figure 3), which makes it possible to reach a wide range of torque, from rest to values above the nominal 20 N.m, according to the motor manufacturer.

Figure 3 show the TIM test bench. It consists of a (1) DC motor/generator VARIMOT BN 132S with rated power of 5.5 kW. The DC motor/generator simulates a load coupled to the shaft of the three-phase induction motor; (2) HBM T40B-200 torque transducer that can operate at speeds of up to 20,000 rpm, up to 200 N.m, accuracy of 0.1 N.m of full scale; (3) bearing bracket used in the alignment of the shafts; (4) WEG W22 Plus three-phase induction motor with a nominal power of 3.7 kW, 380 Vca supply voltage at 60 Hz, 4-pole, and nominal rotation at 1725 rpm. Its function is supply torque to the set; (5) Variable transformer is connected to a bridge rectifier to change the voltage field circuit of the motor/generator. Therefore, DC motor/generator can simulate a variable load.

The TIM can be started by full voltage or through the VFD (model WEG CFW700). With the VFD, it is possible to carry out experiments in different speed and torque ranges, which allows for the control of the speed of the motor shaft through frequency modulation. The test bench DC generator imparts the load to the TIM shaft by electromagnetic braking. The applied torque is controlled by varying the field current. Additionally, the test bench has the data acquisition board NI USB-6215, manufactured by National Instruments, with a 16-bit resolution and a maximum acquisition rate of 250,000 samples per second. Operating speeds were measured using the Minipa digital tachometer, model MDT 2238b, with a resolution of 1 rpm and read accuracy ± 0.05 + 1 digit. Current acquisition was performed using the hall-effect-based linear current sensor ACS712, with a nominal current of 20 A and a total measurement error of approximately 1.5%.

The steps of the algorithm are described as follows (Figure 4): (1) the DAQ NI-USB 6215 data acquisition system uses two phases of motor supply (current signals $i_{a}$ and $i_{b}$) to acquire data at a sampling rate of 30,000 samples per second; (2) then the current instantaneous signals are amplified by multiplication ($i_{a}\times i_{b}$) to improve the SAC-DM’s sensitivity under conditions of the motor operating at variable speed (empirically detected); (3) the result signal ($i_{a}\times i_{b}$) is processed by multiresolution analysis (Wavelet) to eliminate the oscillatory signal component; (4) with the result signal without the oscillatory component, the rate local maxima per second from the signal (SAC-DM) can be calculated; (5) then returns the equation that relates the motor speed with the SAC-DM; (6) parallel to the main signal processing the FFT is used for calibration.

Each phase of the original current signal has a frequency of around 60 Hz (full voltage–utility frequency). However, this value is modified by multiplying these signals, which results in an oscillatory component of approximately 120 Hz. A bank of filters described by the multiresolution analysis (MRA) is used to eliminate the oscillatory signal component.

As will be seen later, through the processing of the TIM current signal, the SAC-DM values vary linearly with the values of the operating speed. With this, the TIM operation speed can be estimated from the chaotic variable, requiring only a short sampling of the original signal. However, when the algorithm is applied to a TIM for the first time, it will be necessary to carry out a calibration process. This process represented in step 5 in Figure 2 consists of a function based on the FFT of the signal, which will estimate the speed for specific TIM operation conditions. In the calibration process, the TIM must have its load slowly varied from a load close to zero to values above or close to the nominal value (in this work, we use a load value 140% higher than the nominal value). During this process, the FFT is calculated for different values and loads while the SAC-DM is calculated. Through FFT, it is possible to obtain the rotation speed on the shaft, which is correlated with the respective calculated SAC-DM value; the relationship between both is linear. The equation of the straight line between the shaft speed value measured by the FFT and the corresponding SAC-DM value provides the calibration function. After this calibration process, the function provides the speed value based on the calculated value from the SAC-DM. The Results Section explains in detail how the calibration process is carried out.

The results obtained by the algorithm are presented in the next section, with the motor operating by direct start and drive through a frequency inverter.

## 8. Results

It is not always possible to obtain a mathematical model or a graphical form from signals obtained through experimental tests that indicate a deterministic series. Time series obtained through the observation of systems can exhibit complex interactions between deterministic and stochastic components [12].

The symbol tree test proposed in [25] was employed in this study to determine whether a time series is deterministic. The symbol tree test involves analyzing a time series segment, in this case, the electrical current signal. If the series is deterministic, the spectra of each partition will cluster and overlap. If the signal is widely scattered and a pattern cannot be observed, the series may be considered stochastic.

Choosing 20 overlapping spectra, as suggested by the previously mentioned authors, tends to be sufficient for determining the similarity of the spectra. Therefore, N was chosen to be 20,000, corresponding to just under one second, considering that a sampling frequency of 30 kHz was used. The partition length was set to l = 1000, and the grouping of “words” was defined as L = 6. With these parameters, the graph has 64 words with 20 overlapping spectra, as observed in Figure 5.

The 0-1 test method proposed in [29] was applied to characterize the electrical current signal from the TIM as chaotic, t. This interactive method provides a direct interpretation of the result. When the data cluster around a value of 0, it suggests that the series does not exhibit chaotic behavior. However, if the values concentrate around a value of 1, this indicates the presence of chaos in the series.

The result of the 0-1 test applied to the TIM electrical current can be seen in Figure 6. It can be seen that the data consistently cluster around a value of 1, with a median of 0.9934. Therefore, it is reasonable to conclude that the TIM electrical current signal can be characterized as chaotic.

### 8.1. Full-Voltage Starting

The results presented in this section refer to the test carried out with the motor driven by the mains voltage. The experiment parameters are shown in Table 3.

The speed values were measured using a digital tachometer, as shown in Table 3. For the experiment, the speed and load range were defined previously. Additionally, the current signals from the two phases of the TIM supply, phase $i_{a}$ and phase $i_{b}$, were obtained. In the first step of the algorithm (as shown in Figure 4), the current signals from the two phases of the motor supply are acquired. For illustrative purposes, the torque range for 0% of the rated load was selected; in Figure 7, item (a) depicts the step in which the two current signals of the TIM are captured. In item (b), these signals are multiplied, and it was empirically discovered that this process substantially amplifies the effect of the chaotic signal. The load close to 0% of the nominal value was used as an example because speed estimation in TIMs presents a challenge in this operating range, considering that the amplitudes of the current signals related to the motor rotation are low.

These signals are multiplied to substantially amplify the chaotic signal effect. The information remains in the multiplied signal, as will be exposed by the calibration block and the MRA responses.

The density of maxima (SAC-DM) calculation is performed after the oscillatory component is eliminated from the signal, as presented in Equation (Equation 6). Figure 8 shows the density of maxima found in the current signal, with the motor operating at 0% of the rated load.

We obtained the response of the chaotic component for each measured speed value after running the presented analysis of the TIM. An interval of 28 s was taken from the signal, counting the peaks every 0.2 s, following Equation (Equation 6), and the result can be seen in Figure 9.

Note that the values obtained from the SAC-DM of the signal shown in Figure 9 have a linear correlation with time. Furthermore, it is possible to observe that, even for little variations in the speed range (in the experiment, the most negligible speed variation was 0.17 Hz), the chaotic component exhibits behavior directly proportional to the motor speed values. The average values μ and the respective standard deviations σX of the SAC-DM are shown in Table 4.

The calibration function estimates the motor speed values for each percentage of the nominal load. The values obtained through the FFT of the signal are shown in Figure 10. It is interesting to observe that, in Figure 10, the energy of the frequency component corresponding to the TIM rotation tends to increase when the rotation moves away from the synchronous speed of the motor.

The high accuracy of the speed values obtained by the FFT of the signal is achieved at the cost of a high time window. For example, the results presented in Figure 10 are achieved by processing a 28 s time window of the signal. Despite this, this step is necessary to provide the speed values associated with the SAC-DM. Table 5 presents a comparison between the speeds obtained by the FFT and the speed values measured with the digital tachometer.

The linear relation between the speed estimated from the FFT and the experimental speed, with the respective correlation coefficient, can be observed in Figure 11.

Similarly, it is possible to see that there is a linear relation between the speed estimated using the FFT and the one estimated using the SAC-DM (see Figure 12). From these results, it is possible to extract Equation (Equation 23), which governs the behavior of the curve:(23)v=0.00079·S+26

In Equation (Equation 23), *v* is the estimation speed, and S is the chaotic component of the SAC-DM.

Figure 13 illustrates a comparative graph between the measured and estimated speeds, through load/speed variation, with the engine operating with a direct start, which allows for a visualization of the curves under dynamic load conditions.

### 8.2. Variable-Frequency Drive (VFD) Starting

Full-voltage starting motor applications become limited due to their narrow operating range. However, Variable-Frequency Drive (VFD) substantially increases the operation speed range of a TIM, allowing for a greater scope of its use in the industrial sector.

This section presents the results of the application of the SAC-DM technique for speed estimation, with the TIM driven by a frequency inverter. The parameters selected for this investigation are shown in Table 6.

Unlike the starting voltage analysis, where the percentage of the rated load is obtained from speed variation, the VFD allows for the modification of the speed by the modulation of the signal. Thus, it is possible to submit each speed value shown in Table 6 to the percentage of the rated load to increase our response dataset. The applied load percentages are 0%, 20%, 40%, 60%, and 80%. This section presents the application of the speed estimation algorithm with the TIM operating at a 0% rated load.

Figure 14 displays the current signals of the TIM when activated by the Variable-Frequency Drive (VFD). As experiments are conducted across various load ranges, the signals are shown for illustrative purposes when the motor operates at 1600 rpm, with 0% of the nominal load. In item (a), the current signals for two phases of the TIM supply, $i_{a}$ and $i_{b}$, are presented, both serving as the starting point for the algorithm. In item (b), the resulting signal from the multiplication between phases $i_{a}$ and $i_{b}$ is displayed.

Similarly, as presented in the previous section, the current signals are multiplied, and then the wavelet transform is applied to the resulting signal.

The window of 0.055 s shows the potential of speed estimation through this technique (Figure 15). Remarkably, despite the significant reduction in the size of the signal data packet to just 1650 samples, the SAC-DM value, which has a frequency of 8600 Hz, remains consistent with the averages computed over larger time windows.

From the results of the SAC-DM of each speed value, it is possible to observe (Figure 16) the distribution of the chaotic component as a function of time.

It is possible to observe again that the linear and constant correlations of the SAC-DM are associated with each speed as a function of time, even for the significantly more comprehensive speed range. Table 7 shows the average SAC-DM and the standard deviation associated with the observed speed. The generated calibration function is shown in Figure 17.

The speed values measured with the tachometer and those estimated by the FFT are shown in Table 8, and a graphical representation can be seen in Figure 18.

A graphical representation of Table 8 can be seen in Figure 18.

Finally, a trend curve is produced, depicting the distribution of the SAC-DM values alongside their corresponding speeds, as illustrated in Figure 19.

The function that determines the variation in the SAC-DM as a function of speed for this experiment is as follows:(24)v=−0.007·S+86

## 9. Conclusions

This paper presents the development of an algorithm as a method for estimating the shaft speed of a three-phase induction motor (TIM), with a Signal Analysis based on Chaos using Density of Maxima (SAC-DM). The algorithm brings the advantages presented in the literature of a non-invasive technique, thus reducing the need for equipment for its implementation. The results indicate the potential of the technique for estimating the speed of a TIM, predominantly when the motor operates under a dynamic load, capable of detecting a narrow range of speed variations of up to 0.167 Hz (10 rpm).

It should be noted that, while techniques based on the FFT of the signal for speed estimation are highly accurate, they are also limited by the requirement for stationary operation of the TIM and the associated high computational effort. In contrast, the SAC-DM technique offers a much lower time window (0.2 s window—6000 samples) when compared to estimates obtained through the FFT (28 s—840,000 samples). Additionally, for TIMs starting via VFD, the speed estimation time can be further reduced to 0.055 s.

## Figures and Tables

**Figure 1 entropy-26-00361-f001:**
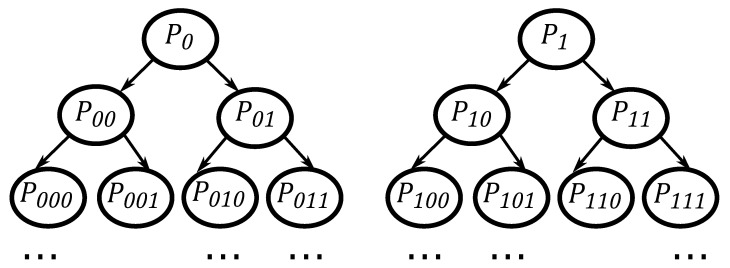
Symbol tree. Adapted from [24].

**Figure 2 entropy-26-00361-f002:**
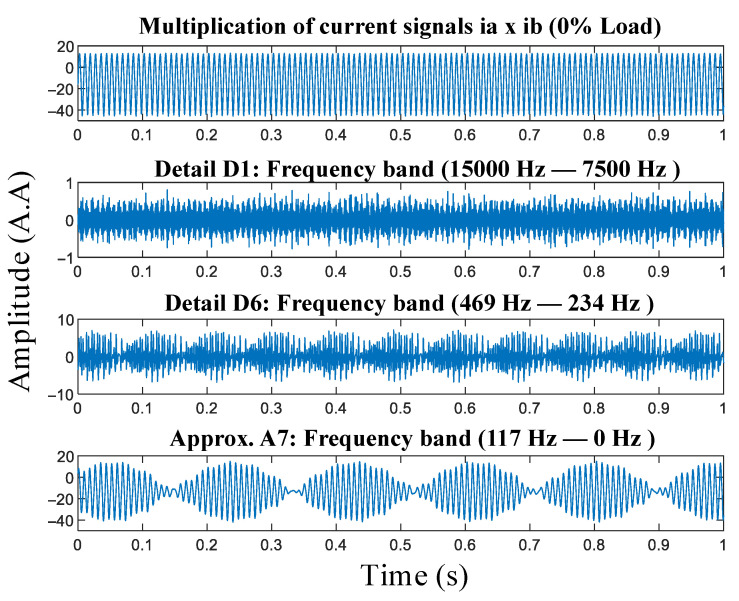
Decomposition by MRA.

**Figure 3 entropy-26-00361-f003:**
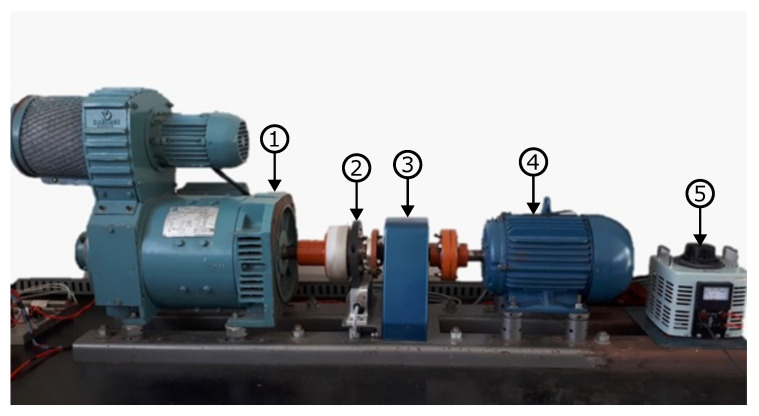
Motor test bench used in experiments.

**Figure 4 entropy-26-00361-f004:**
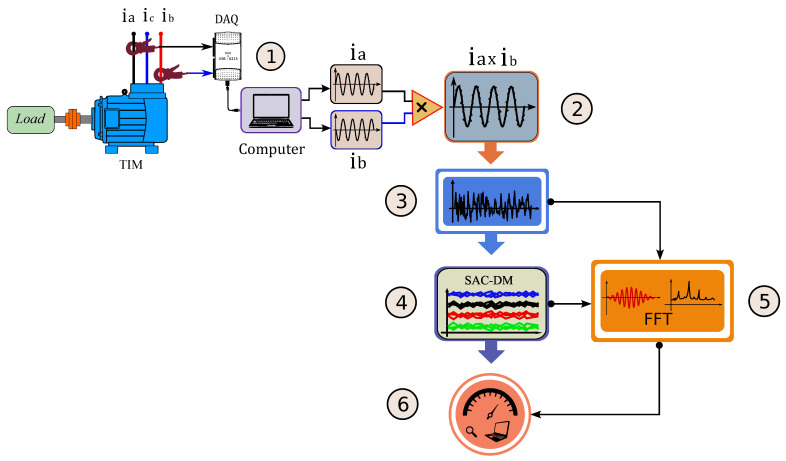
Representation of the proposed algorithm.

**Figure 5 entropy-26-00361-f005:**
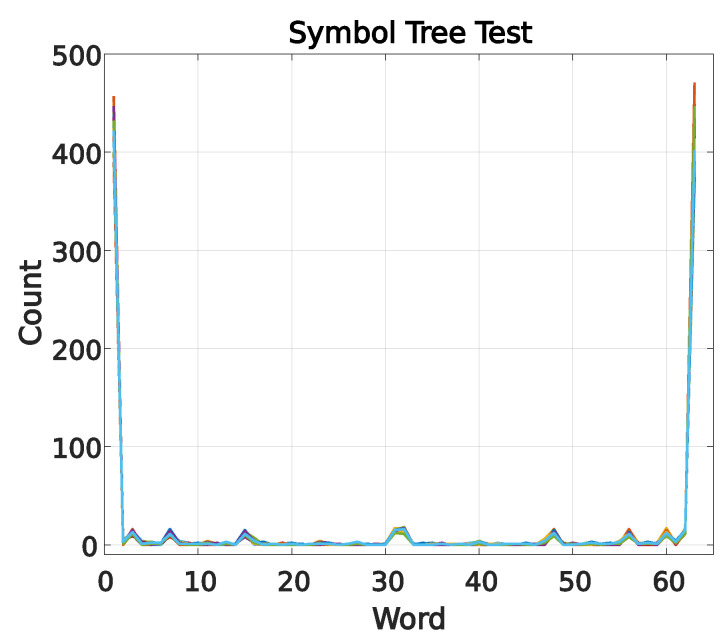
Symbol tree test for the TIM current signal.

**Figure 6 entropy-26-00361-f006:**
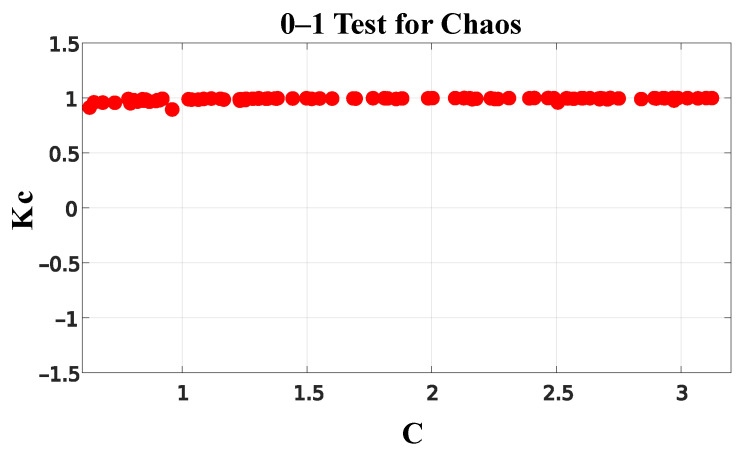
The 0–1 test for TIM electrical current.

**Figure 7 entropy-26-00361-f007:**
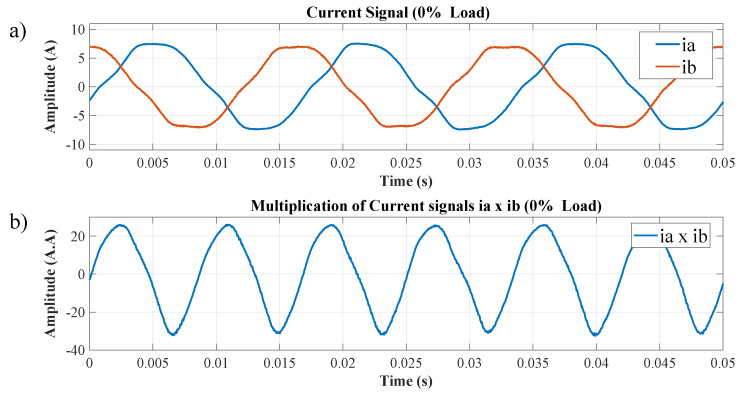
(**a**) Current signals of two phases of the TIM, (**b**) result of multiplied TIM phase current signals.

**Figure 8 entropy-26-00361-f008:**
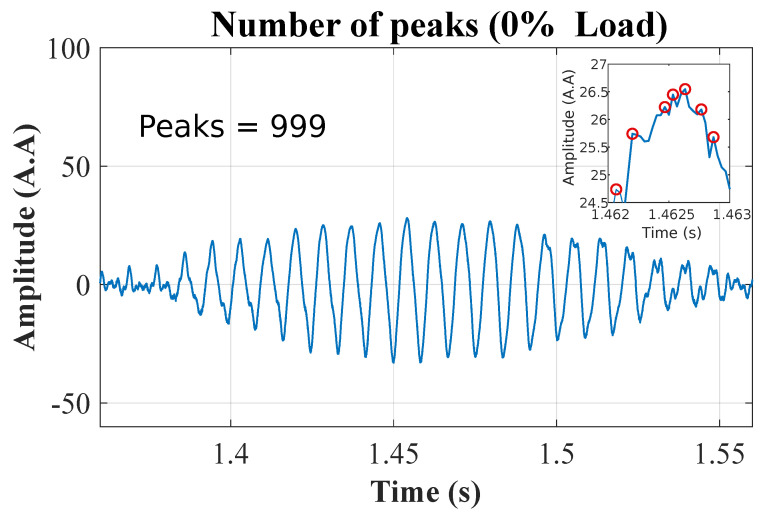
The resulting signal from the MRA, with the peaks highlighted within the range. The total number of peaks is 999.

**Figure 9 entropy-26-00361-f009:**
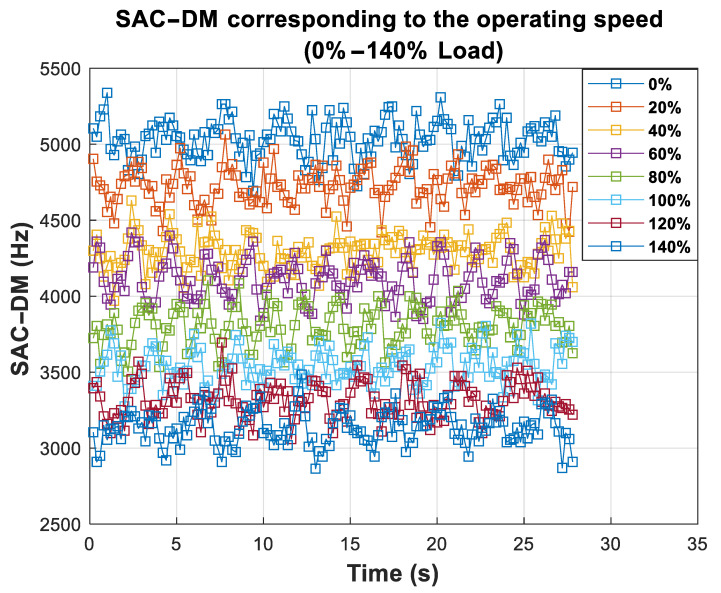
Response of SAC-DM values for different percentages of the rated load.

**Figure 10 entropy-26-00361-f010:**
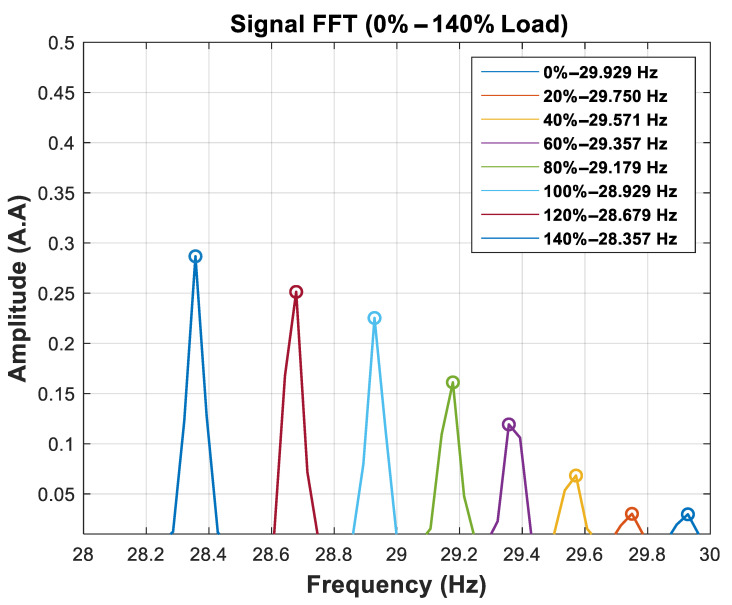
Calibration function.

**Figure 11 entropy-26-00361-f011:**
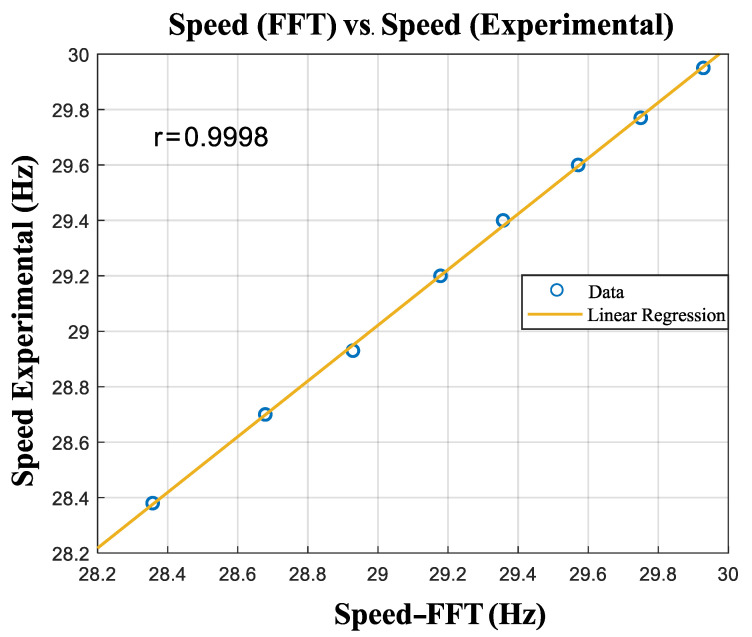
Linear association between the motor speed and the speed obtained by the calibration function.

**Figure 12 entropy-26-00361-f012:**
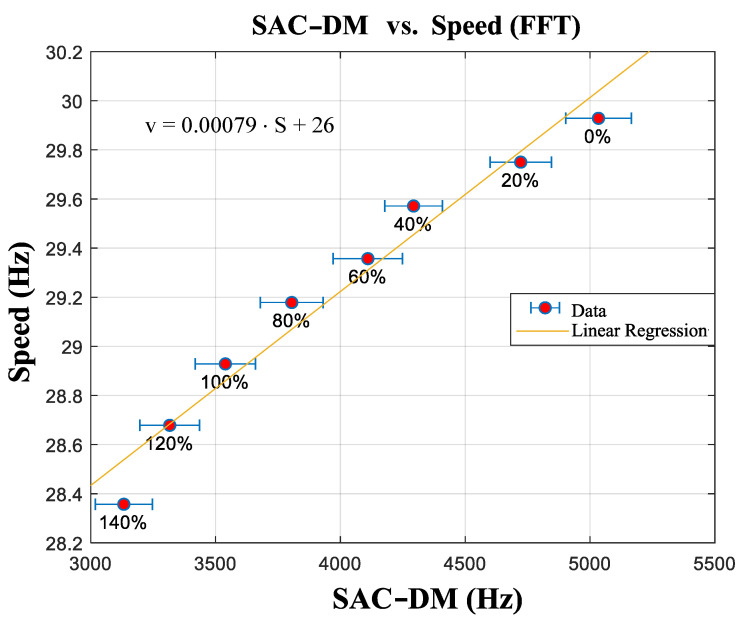
Linear association between the SAC-DM chaotic component and the estimated speed (FFT).

**Figure 13 entropy-26-00361-f013:**
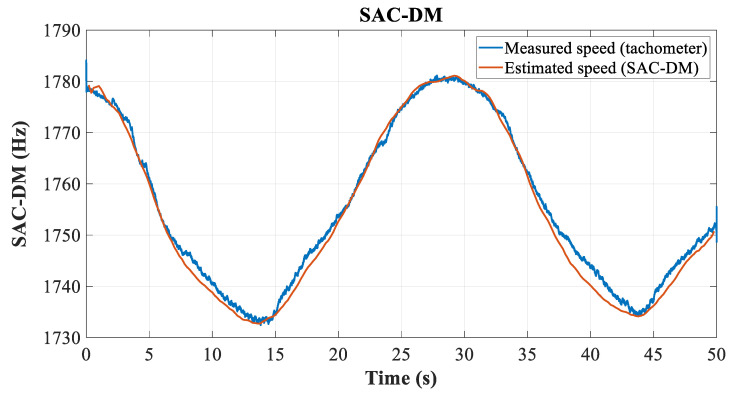
Comparative graph between estimated and measured speeds under variable load conditions.

**Figure 14 entropy-26-00361-f014:**
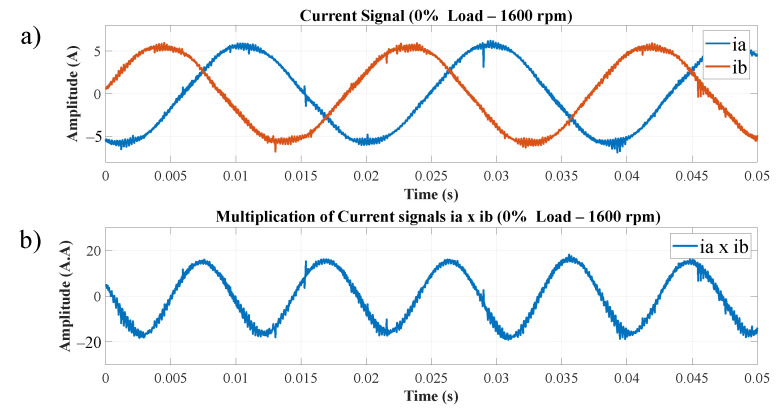
(**a**) Current signals of two phases of the TIM (VFD), (**b**) signal multiplied ia x ib for 0% rated load and 1600 rpm (VFD).

**Figure 15 entropy-26-00361-f015:**
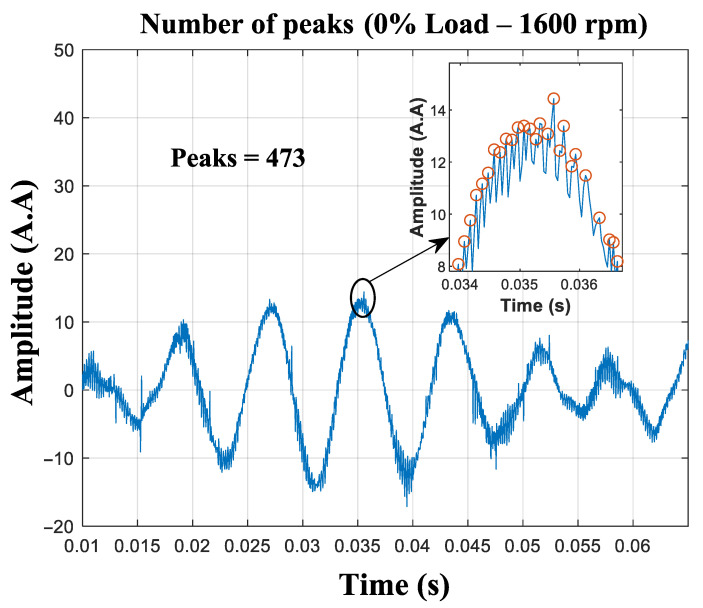
The MRA of the signal, with the peaks highlighted (0% load—1600 rpm).

**Figure 16 entropy-26-00361-f016:**
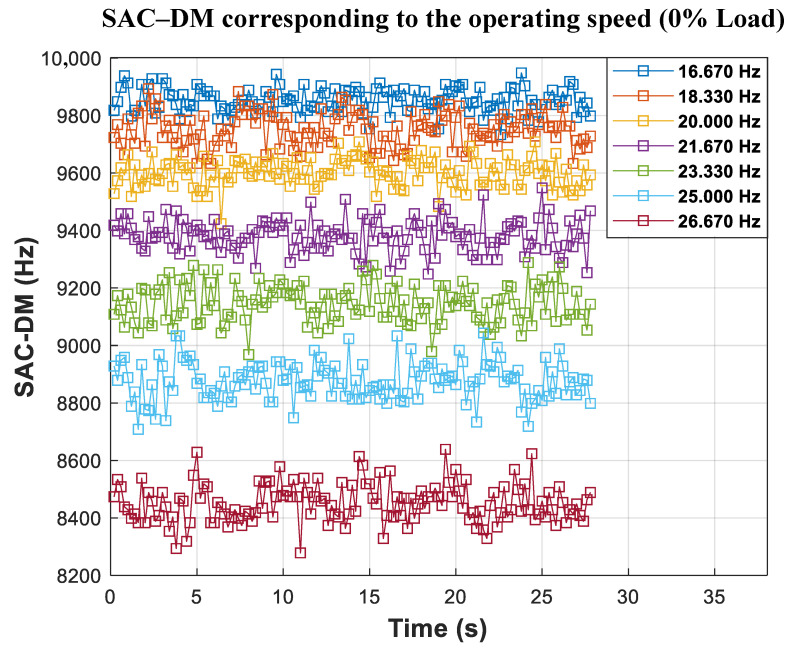
SAC-DM values for each speed value associated with 0% rated load for variable-frequency drive (VFD) application.

**Figure 17 entropy-26-00361-f017:**
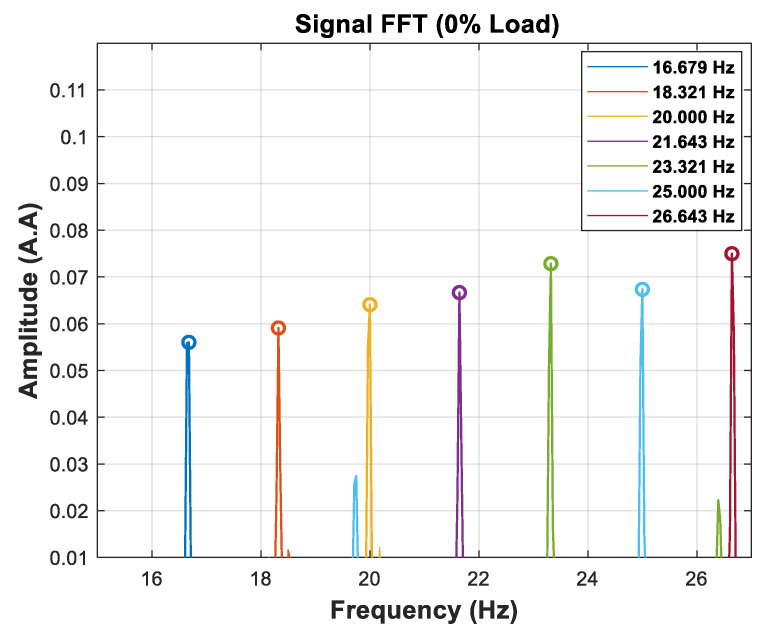
Calibration function for 0% of rated load.

**Figure 18 entropy-26-00361-f018:**
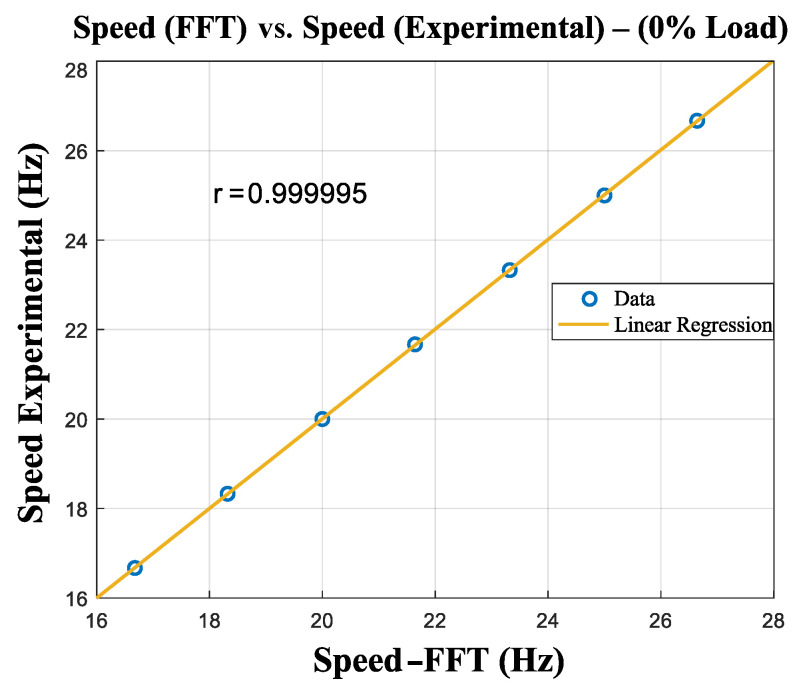
Linear association between the speed estimated by the calibration function and the speed obtained by the digital tachometer (0% load).

**Figure 19 entropy-26-00361-f019:**
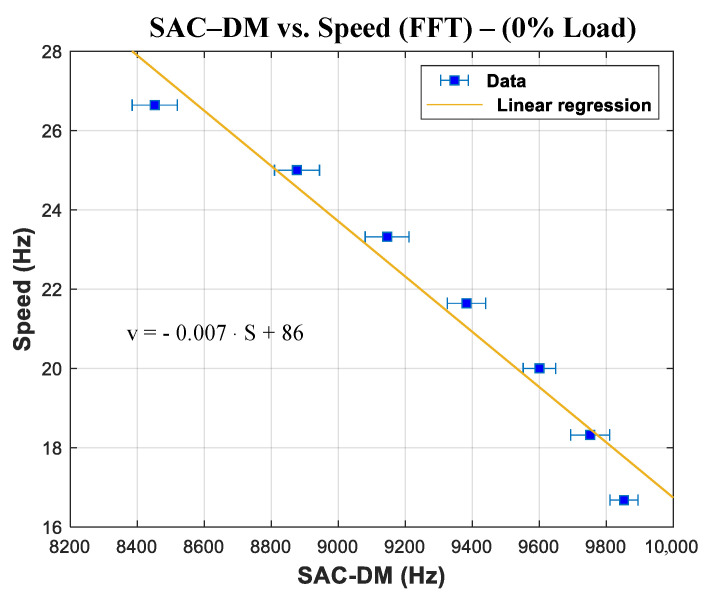
Linear association between the SAC-DM chaotic component and the estimated speed (FFT) with 0% load.

**Table 1 entropy-26-00361-t001:** Comparison between the main techniques for induction motors speed estimation.

Work	Domain	Technique	Response Time	Relative Error
[13]	Frequency	Chirp—z transform	0.2 s	<0.16%
[14]	Time	Mathematical motor model	1.2 s	-
[15]	Frequency	Modified Prony Method	0.3 s	0.15%
[16]	Frequency	Adaptive Sliding Window	0.1 s	1%
[17]	Time	Rotor flux second derivative	0.05 s	-
[18]	Time	Mathematical motor model	1 s	1%
[7]	Frequency	Hilbert Transform and Goertzel Algorithm	1 s	<0.15%
[19]	Time	TIM equivalent circuit	-	-
[20]	Frequency	FFT	-	0.9%
[3]	Time	Reactive-power-based model reference	-	0.5% to 1.5%
[21]	Frequency	Sliding-mode observer (SMO)	-	-
Proposed Article	Time/ Frequency	Chaos theory (SAC-DM)	0.2 s	0.31%

**Table 2 entropy-26-00361-t002:** Frequency window according to the MRA component.

MRA Component	Frequency Window (Hz)
D1	15,000–7500
D2	7500–3750
D3	3750–1875
D4	1875–937.5
D5	937.5–468.7
D6	468.7–234.4
D7	234.4–117.2
A7	117.2–0

**Table 3 entropy-26-00361-t003:** Test configuration for full-voltage starting TIM for 840,001 samples at 30,000 samples per second.

Torque (N.m)	Load (%)	Speed (rpm)	Speed (Hz)
0	0	1797	29.95
4	20	1786	29.77
8	40	1776	29.60
12	60	1764	29.40
16	80	1752	29.20
20	100	1736	28.93
24	120	1722	28.70
28	140	1703	28.38

**Table 4 entropy-26-00361-t004:** Average and standard deviation of SAC-DM values for each speed.

Load (%)	Speed (RPM)	Speed (Hz)	μ (SAC-DM)	σX (SAC-DM)
0	1797	29.95	5033.801	131.4051
20	1786	29.77	4722.342	122.8576
40	1776	29.60	4293.133	115.2456
60	1764	29.40	4109.962	139.0379
80	1752	29.20	3805.229	125.9275
100	1736	28.93	3539.302	120.6895
120	1722	28.70	3316.498	119.6956
140	1703	28.38	3132.571	114.5949

**Table 5 entropy-26-00361-t005:** Comparison of the speed values of the experiment and the values obtained by the signal FFT.

Speed (Hz)	Speed FFT (Hz)	Relative Error
29.95	29.92857	0.0715%
29.77	29.75000	0.0672%
29.60	29.57143	0.0965%
29.40	29.35714	0.1460%
29.20	29.17857	0.0734%
28.93	28.92857	0.0049%
28.70	28.67857	0.0747%
28.38	28.35714	0.0805%

**Table 6 entropy-26-00361-t006:** Test parameters (VFD) for 84,001 samples at 30,000 samples per second.

Speed (rpm)	Speed (Hz)
1600	26.667
1500	25.000
1400	23.333
1300	21.667
1200	20.000
1100	18.333
1000	16.667

**Table 7 entropy-26-00361-t007:** Average and standard deviation of SAC-DM values for each speed (0% load) for VFD application.

Speed (rpm)	Speed (Hz)	μ (SAC-DM)	σX (SAC-DM)
1000	16.667	9852.926	41.776
1100	18.333	9751.972	58.458
1200	20.000	9600.342	48.456
1300	21.667	9382.681	57.294
1400	23.333	9145.814	65.509
1500	25.000	8876.758	67.068
1600	26.667	8452.009	67.292

**Table 8 entropy-26-00361-t008:** Comparison of experimental speed values with those obtained by the signal FFT (0% load).

Speed [Tachometer] (Hz)	Speed [FFT] (Hz)	Relative Error (%)
16.667	16.679	−0.069
18.333	18.321	0.0631
20.000	20.000	0.000
21.667	21.643	0.1114
23.333	23.321	0.0496
25.000	25.000	0.000
26.667	26.643	0.0905

## Data Availability

The data presented in this study are available on request from the corresponding author.

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
