# Peer review of "Sensorless Speed Estimation of Induction Motors through Signal Analysis Based on Chaos Using Density of Maxima"

_entropy, 2024, doi:10.3390/e26050361_

Round 1

Reviewer 1 Report

Comments and Suggestions for Authors

Author Response

All the reviews are attached.

Reviewer 2 Report

Comments and Suggestions for Authors

This paper presents an algorithm for speed estimation in three-phase induction motors using the chaotic variable of maximum density. The technique used in this work analyzes the current signals from the motor power supply without invasive sensors on its structure. The results show that the speed estimation is achieved with a response time lower than those obtained by classical techniques based on the Fourier Transform.

The authors present a decent research work. I think the paper is suitable to be published after minor revision. Several suggestions and questions are listed as follows:

1) I notice the rotational speed of the TIM is in the range of 0-1600 rpm. Did the authors try a higher speed? Could the proposed method be applied to estimate a high-speed motor, like 6000 rpm or more than 10000 rpm?

2) The quality of the figures should be improved as the image pixels are relatively low

Author Response

All the reviews are attached.

Reviewer 3 Report

Comments and Suggestions for Authors

The manuscript presents an interesting contribution to the non-intrusive speed estimation of induction motors. It is common for the motor shaft to be inaccessible, making speed measurement difficult and often costly. This work provides a method aimed at facilitating speed measurement and presents intriguing results. However, there is room for improvement in the manuscript.

Sections 3, 4, 5 should be further developed and the mathematical tool used should be explained in more detail. A notation table should be included.

The location of table 1 is not correct, it is in section 3 and belongs to the state of the art. This is a general problem of the manuscript; the figures are not usually close to the place where they are cited.

In line 90 the authors indicate that q(t) would correspond to the “armature current”. It is not clear to me whether this would be the product of stator current phase a and b (iaxib). Line 92, q(t)? or qi(t) ? Clarify

Methodology

The method employs the measurement of the current in phases A and B of the motor supply. The authors should specify the error in the current measurement. It is important to determine the cost of acquiring equipment with the necessary functionalities to implement the measurement. It should be noted that the authors aim for an affordable measurement method. Figures 7 and 8 should be placed before Table 3 since they are referenced earlier in the text. For clarity, Table 3 should include the load percentage (%). Authors chose 0% load as an example to present the results, but it would be beneficial to include an intermediate load state, such as 60%, and redo Figures 7, 8, and 9, as no-load is not representative. Figure 11 should precede Table 4, as it is mentioned earlier in the text. Figures 15 and 16 currently depict the no-load condition. It would be better to represent a load of 60%, for example. Before Table 3, Figures 8, 9, and 10 should be placed since they are cited earlier in the text.

Conclusion

Again, it should shown a load example in Figure 20 and 21.

Author Response

All the reviews are attached.

Reviewer 4 Report

Comments and Suggestions for Authors

1. It is fundamental to correct orthography, grammatical and format errors, the text it is not
clear enough. The sections are read independently and do not appear to be related
between them. There are some format errors along the document (some examples are in
lines 140, 144, 145, etc.).
2. In line 52 authors mention “chaotic signals have immunity to background noise”, that
affirmation is not true at all, it is not a generalized characteristic of chaos.
3. In section 2 some contributions of the presented work are mentioned, in lines 79-82
authors mention that this methodology has shorter times and lower computational cost.
Please add a reference to support that or did the authors estimate the computational cost
to evaluate the performance of the methods?
4. Equation 1 seems to be incorrect.
5. Table 1 it is not explained clearly.
6. The text says in lines 172 and 173: “Equation (4) shows the expression for the mean
square displacement in terms of the conversion variables:”. But the expression it is not the
correct it should be (12)
7. In eq. (13) V c and V OSC are not defined
8. In line 179 what does it mean 0asn
9. In eq. 14 there are not parentheses for cosines
10. In eq. 15 E[Ï•] is not defined (Is this variable called expectation?), In section 5 there are
variables that there are not defined.
11. In section 6 authors mentioned that the used work bench could be inferred by the
provided information, but it is not true, it is something that couldn’t be inferred.
12. In line 208 D.C. Acronym is used, but in the rest of the document is used DC, please unify
the acronyms. Some acronyms are not defined please define all the acronyms you use.
13. Why authors use two phases (i a and i b )?
14. In section 6 In line 219 authors mention they use Multiple resolution analysis, please
explain all the used methods and theory in only one section.
15. Authors mention that they use auxiliary methods like FFT and Multiple resolution analysis
which introduce additional complexity to the proposed methodology, based on that
authors should explain better why they said that this technique has shorter response
times and lower computational cost. Better response times and better approximations
(lower relative error) with other techniques are shown in table 1.
16. In line 232 why authors select that frequency range (0Hz to 234Hz)?
17. Figure 4 is not explained, it does not have labels at the axis, why did you select those
frequency bands?
18. The calibration process is not clear, authors should include the mathematical process or a
reference.
19. In line 254, mention the name of the method.
20. In section 7 there are not defined acronyms.
21. Authors mention that signals are multiplied to amplify the chaotic effect, could you
explain why chaos increases based on that? This sentence “These signals are multiplied to
amplify the chaotic signal effect substantially” should be demonstrated or cited.
22. In line 287 authors mention equation 7 but the reference seems to be incorrect, same in
line 291.

23. The relationship between speed and SAC-DM is not clear enough, please add the speed
estimation equation mentioned in line 337.
24. In general, images and tables should be explained concisely and clearly.
25. In general, images quality should be improved.

Author Response

All the reviews are attached.

Round 2

Reviewer 3 Report

Comments and Suggestions for Authors

In my opinion, the authors have made improvements to the manuscript. However, I cannot accept it for publication until all comments from the previous review have been fully addressed.

Comment 1: It is crucial to determine the cost of acquiring equipment with the necessary functionalities to implement the measurement. The authors sidestep this question by stating that these devices are widely available on the market. I believe it would enrich the paper if they provided an approximate cost of implementing their proposed speed measurement method.

Comment 2: Address and represent the speed calculation for load states other than no-load. The authors have avoided testing the induction motor under variable load conditions (variable speed). Testing the method solely under a no-load regimen is inadequate for the manuscript to be accepted for publication. The method should be evaluated across a broad range of speeds. It is pertinent to mention that, although the title of the manuscript is 'Sensorless Speed Estimation...', only near-synchronous speed is actually estimated.